# Impact of Theaflavins-Enriched Tea Leaf Extract TY-1 against Surrogate Viruses of Human Norovirus: In Vitro Virucidal Study

**DOI:** 10.3390/pathogens11050533

**Published:** 2022-05-02

**Authors:** Israa M. A. Mohamed, Dulamjav Jamsransuren, Sachiko Matsuda, Haruko Ogawa, Yohei Takeda

**Affiliations:** 1Graduate School of Animal and Veterinary Sciences and Agriculture, Obihiro University of Agriculture and Veterinary Medicine, 2-11 Inada, Obihiro 080-8555, Japan; israahygiene@gmail.com; 2Department of Animal and Poultry Hygiene & Environmental Sanitation, Faculty of Veterinary Medicine, Assiut University, Assiut 71526, Egypt; 3Department of Veterinary Medicine, Obihiro University of Agriculture and Veterinary Medicine, 2-11 Inada, Obihiro 080-8555, Japan; duuya.dj@gmail.com (D.J.); chaka@obihiro.ac.jp (S.M.); hogawa@obihiro.ac.jp (H.O.); 4Research Center for Global Agromedicine, Obihiro University of Agriculture and Veterinary Medicine, 2-11 Inada, Obihiro 080-8555, Japan

**Keywords:** feline calicivirus, murine norovirus, theaflavins, catechins, virucidal, disinfection

## Abstract

Using an effective natural virucidal substance may be a feasible approach for preventing food-borne viral contamination. Here, the virucidal efficacy of theaflavins (TFs)-enriched tea leaf extract (TY-1) against feline calicivirus (FCV) and murine norovirus (MNV), surrogates of human norovirus (HuNoV), was evaluated. The virus solutions were mixed with various dosages of TY-1 and incubated at 25 °C for different contact times. TY-1 reduced the viral titer of both surrogate viruses in a time- and dosage-dependent manner. A statistically significant reduction in the viral titer of FCV by 5.0 mg/mL TY-1 and MNV by 25.0 mg/mL TY-1 was observed in 10 s and 1 min, respectively. Furthermore, TY-1 reduced the viral titer of FCV and MNV on the dry surface in 10 min. The multiple compounds in TY-1, including TFs and catechins, contributed to its overall virucidal activity. Furthermore, the effect of TY-1 on viral proteins and genome was analyzed using Western blotting, RT-PCR, and transmission electron microscopy. TY-1 was found to promote the profound disruption of virion structures, including the capsid proteins and genome. Our finding demonstrates the potential of using TY-1 as a nature-derived disinfectant in food processing facilities and healthcare settings to reduce viral load and HuNoV transmission.

## 1. Introduction

Infection by food-borne enteric-related viruses is the main reason for acute gastroenteritis in humans [1]. Among the food-borne viruses, human norovirus (HuNoV), a non-enveloped RNA virus of the family *Caliciviridae*, was first recognized during an acute gastroenteritis outbreak in Norwalk, OH, USA, in 1968 [2]. Worldwide, HuNoV accounts for about one-fifth of all acute gastroenteritis cases among all age groups, leading to over 200,000 deaths per year in both high- and low-income countries, and causing large economic costs for healthcare systems amounting to approximately USD 4.2 billion every year [1,3,4,5]. HuNoV likely maintains its infectivity on human hands, inanimate objects, and environmental surfaces for a long time. For example, HuNoV RNA can be detected on human finger pads after 2 h of incubation [6] and on environmental surfaces after ≥28 days [7,8]. This feature of the virus probably contributes to the high prevalence of outbreaks observed in closed environments, such as long-term care facilities, cruise ships, schools, and food service establishments [9,10]. Currently, there are no vaccines or antiviral drugs approved for HuNoV. Meanwhile, multiple virucidal chemical compounds for the disinfection against the surrogate viruses of HuNoV have been reported on environmental surfaces [11,12,13]. However, the adverse health effects of some of these virucidal chemical compounds and their residual property in the environment are disadvantageous for their application as daily-use disinfectants. Therefore, there is a need to develop an efficient, harmless, and environmentally friendly strategy to inactivate HuNoV and prevent its infection.

One major issue hindering the research on HuNoV is the difficulty in virus propagation in major cell culture systems. In some recent studies, HuNoV was successfully cultivated using B cells, human enteroids, and zebrafish larvae [14,15,16]. However, these methods of virus cultivation have not been widely employed because of some limitations (e.g., complicated procedures, low reproducibility, or low virus amplification efficacy). Owing to its high specificity and sensitivity, the reverse transcription–quantitative polymerase chain reaction (RT-qPCR) assay is considered the gold standard for assessing and detecting the viral RNA of HuNoV in environmental and clinical samples [17]. Nevertheless, such genome-based molecular approaches cannot be used to distinguish between infectious and noninfectious viruses. Altogether, the lack of a well-established HuNoV cell culture model has prevented the assessment of the virus infectivity via the 50% tissue culture infectious dose (TCID_50_) or plaque assays, which can be regularly applied to surrogate viruses. Accordingly, the evaluation of the HuNoV inactivation efficiency of disinfectants continues to largely rely on the use of easily cultivable surrogates with relatively close structural and genetic similarities to HuNoV, such as the feline calicivirus (FCV), murine norovirus (MNV), and Tulane virus [18,19,20,21]. FCV is a member of the *Vesivirus* genus, which causes a respiratory disease in domestic cats. While the disease symptoms are different from those of HuNoV, FCV has been used as an HuNoV surrogate virus for a long time, and it was approved as such by the U.S. Environmental Protection Agency to evaluate the efficacy of virucidal agents [22]. MNV was first recognized in 2003 when researchers detected sporadic deaths in severely immunocompromised mice due to it [23]. Additional characterization revealed MNV as a novel member of the *Norovirus* genus [20], which has structural and genetic features that are similar to those of HuNoV. Consequently, both FCV and MNV have been frequently used as surrogate viruses of HuNoV in virus inactivation studies [24,25,26,27,28].

Using an effective nature-derived substance has been proposed as a feasible approach. In ancient times, medicinal plants enriched with flavonoids, polyphenolic compounds, and alkaloids were found to have antimicrobial, antiseptic, and microbe-inactivating activities, and thus were used to eliminate pathogenic microbes. Since many herbal plants are harmless to humans as well as the environment, their potential use as hand disinfectants against pathogens has become a research focus [29,30,31]. Thus, these herbal plants can be tested for their ability to prevent HuNoV infection. Previously, we reported that theaflavins (TFs)-enriched tea extract named TY-1, extracted from the leaves of raw green tea [32] and containing other polyphenols including catechins, exhibited virucidal activities against enveloped RNA viruses, such as severe acute respiratory syndrome coronavirus 2 (SARS-CoV-2) and influenza A virus (IAV) [33,34]. In this study, the virucidal effect of TY-1 on non-enveloped RNA viruses, FCV and MNV, was evaluated. In addition, the mechanism of TY-1 was investigated, and the possible application of TY-1 for preventing HuNoV infection is discussed.

## 2. Results

### 2.1. Virucidal Effects of TY-1 on FCV and MNV

First, the viral titers of phosphate-buffered saline (PBS)- and 2.5 mg/mL dextrin (Dex)-treated FCV and MNV were compared. The viral titer in the Dex group was comparable to that in the PBS group at both 0 and 24 h contact times (Appendix A), and the results suggested that Dex does not possess virucidal activity; therefore, it can be used as a solvent control of TY-1. The virucidal efficiency of TY-1 against FCV and MNV at various dosages was evaluated. TY-1 exhibited a time- and dosage-dependent inactivating activity against both viruses. Specifically, at the contact times of 1 min to 24 h, the viral titers of the FCV solution treated with 1.3, 2.5, and 5.0 mg/mL TY-1 were significantly decreased compared with that of the FCV solution treated with Dex. Moreover, at 6 h and 24 h contact times, the viral titers of the solution treated with 2.5 and 5.0 mg/mL TY-1 were below the detection limit; the reductions in viral titer by 2.5 and 5.0 mg/mL TY-1 were ≥3.8 log_10_ TCID_50_/mL compared with the Dex group at 6 h. At the 10 s contact time, the significant reductions in viral titer in the solutions treated with 2.5 and 5.0 mg/mL TY-1 were 0.5 and 1.3 log_10_ TCID_50_/mL, respectively (Figure 1A). Furthermore, at the contact times of 1–24 h, the viral titers of the MNV solution treated with 0.3–5.0 mg/mL TY-1 were significantly reduced compared with that treated with Dex. At the 24 h contact time, the viral titer of the MNV solution treated with 5.0 mg/mL TY-1 was lower than the detection limit (reduction by ≥2.7 log_10_ TCID_50_/mL). Additionally, at the 10 min contact time, the treatment with 5.0 mg/mL TY-1 caused a significant reduction in the viral titer by 0.5 log_10_ TCID_50_/mL compared with the Dex group (Figure 1B). The dosages of TY-1 that did not show a significant reduction in the viral titer at certain reaction times were not tested for the subsequent shorter contact times. Subsequently, the virucidal activities of the higher dosage (25.0 mg/mL) of TY-1 against both the viruses were evaluated at a contact time of 1 min. As a result, the significant reductions in the viral titer of FCV and MNV treated with 25.0 mg/mL TY-1 were ≥3.44 and 0.88 log_10_ TCID_50_/mL, respectively (Figure 1C,D).

Furthermore, we evaluated the virucidal impact of 0.034 mg/mL catechins, 0.083 mg/mL TFs, and 0.116 mg/mL TFs+catechins against FCV and MNV compared with that of 5.0 mg/mL TY-1. The concentrations of catechins, TFs, and TFs+catechins were equivalent to their concentrations in the 5.0 mg/mL TY-1 solution. In the case of FCV, at the 3 h contact time, the reduction in viral titer by TFs, TFs+catechins, and TY-1 were 3.1, 3.0, and ≥4.3 (below the detection limit) log_10_ TCID_50_/mL, respectively; there was no statistical difference in the viral titers between the treatments with catechins and Dex. At the 24 h contact time, the reduction in viral titer by catechins, TFs, TFs+catechins, and TY-1 were 0.9, 3.3, 3.7, and ≥3.3 (below the detection limit) log_10_ TCID_50_/mL, respectively (Figure 2A). In the case of MNV, at the 3 h contact time, the reduction in viral titer by catechins, TFs, TFs+catechins, and TY-1 were 1.4, 2.1, 2.5, and 2.1 log_10_ TCID_50_/mL, respectively. At the 24 h contact time, the reduction in viral titer by catechins, TFs, TFs+catechins, and TY-1 were 1.1, 2.4, 2.7, and ≥2.4 (below the detection limit) log_10_ TCID_50_/mL, respectively (Figure 2B).

### 2.2. Virucidal Effect of TY-1 on FCV and MNV on a Dry Surface

The virucidal efficiency of TY-1 against FCV and MNV on a dry surface was evaluated. Specifically, the reductions of the viral titers of FCV and MNV by 5.0 mg/mL TY-1 were 2.75 and 1.5 log_10_ TCID_50_/mL at the contact time of 10 min, respectively (Figure 3A,B).

### 2.3. Effect of TY-1 on MNV Structural Protein

A solution with purified MNV, which was used to eliminate the effect of non-viral proteins, was mixed with the Dex or TY-1 solution. Afterward, the effect of TY-1 on the MNV VP1 protein was evaluated using Western blotting. There was no change in the protein band patterns of the Dex- and TY-1-treated viruses at the contact time of 0 h. Meanwhile, at the 24 h contact time, the VP1 band was detected in the Dex-treated virus solution but not in the TY-1-treated virus solution (Figure 4).

### 2.4. Effect of TY-1 on FCV and MNV Genomes

The effect of TY-1 on the genome of purified FCV and MNV at the contact times of 0 and 24 h was evaluated using RT-PCR. There was no difference in the band intensities of the virus-specific PCR products of the Dex-treated and TY-1-treated viruses at 0 h contact time. In contrast, while PCR products were detected in Dex-treated viruses, they were scarcely detected in TY-1-treated viruses after 24 h (Figure 5A,B).

### 2.5. Effect of TY-1 on FCV and MNV Viral Particles

The morphological changes in FCV and MNV virions induced by 5.0 mg/mL TY-1 were analyzed using transmission electron microscopy (TEM). At the 6 h contact time, multiple FCV and MNV virions appeared to maintain normal structures in the Dex-treated solution (Figure 6A,C and Appendix A). However, many TY-1-treated FCV and MNV virions showed abnormal structures (e.g., decrease in diameter of viral particles, disappearance of cup-shaped depressions on the capsid), and the aggregation of viral particles was observed. Furthermore, the number of intact viruses was reduced in response to the TY-1 treatment (Figure 6B,D and Appendix A).

## 3. Discussion

Given the deficiency of a major cell cultivation system for HuNoV research, a system using cultivable surrogate viruses is proposed. Accordingly, the development of virucidal materials became essential to reduce food-borne HuNoV outbreaks. Sodium hypochlorite (NaClO) is a recommended disinfectant for HuNoV. Duizer et al. [18] reported that sufficient dosages (around 100–3000 ppm) of NaClO exhibited virucidal activities against FCV and MNV, both in solution and on surfaces, within a few minutes, and that the degree of virus inactivation was proportional to the dosage of NaClO. Despite its broad and rapid virucidal activity, the main drawback of using NaClO is that it is not always approved or appropriate for application on environmental and food surfaces because of its toxicity and corrosiveness. On the other hand, there have been many reports demonstrating the virucidal effects of plant extracts [35,36].

Previously, we found that TFs-enriched tea leaf extract, TY-1, also containing other polyphenols such as catechins, showed multiple modes of virucidal activity against enveloped viruses, such as SARS-CoV-2 and IAV [33,34]. In this study, we evaluated the virucidal effect of TY-1 on non-enveloped HuNoV surrogates, FCV and MNV. Similar to our findings regarding the targeting of enveloped viruses by TY-1, TY-1 exhibited time- and dosage-dependent virucidal efficiency against FCV and MNV. We observed a potent and rapid TY-1-mediated inactivation of FCV compared with MNV, where 2.5 mg/mL TY-1 inactivated FCV in 10 s, while 5.0 mg/mL TY-1 took more than 10 min for the inactivation of MNV (Figure 1A,B). However, a higher dosage (25.0 mg/mL) of TY-1 produced a statistically significant virucidal activity even against MNV at the contact time of 1 min (Figure 1D). Furthermore, on a dry surface, FCV was more sensitive to TY-1 compared with MNV (Figure 3). These results correspond with the previous finding that FCV is more sensitive than MNV to plant-derived polyphenols, likely related to the relative instability of FCV compared with MNV [37,38,39]. Ueda et al. [40] tested the virucidal ability of green tea extract against FCV and MNV and reported that both viruses were inactivated by the extract at the contact time of 3 min, with FCV being more susceptible to the virucidal effect of the extract than MNV. Another study reported that a chitosan-based film containing ≥10% of green tea extract inactivated MNV solutions at the contact time of 3 h [41]. Additionally, 10 mg/mL green tea extract recorded 1.42 and 1.96 log_10_ TCID_50_/mL reduction in viral titer of MNV on clean stainless steel and glass surfaces, respectively, at the contact time of 15 min [42]. In these studies, the green tea extract showed time- and dosage-dependent virucidal activities, which were similarly observed in the present study.

We also investigated the mode of action of TY-1 against FCV and MNV. According to the Western blotting analysis, the band of the VP1 capsid protein was disappeared in the TY-1-treated MNV (Figure 4), consistent with the previous observation that *Saxifraga spinulosa*, a flavonoid-rich medicinal herb, affected the viral capsids and structural proteins of enveloped and non-enveloped viruses [43]. Notably, in non-enveloped viruses, the capsid helps protect the viral RNA and establish infection through adsorption to the host cell [44]. Hence, the effect of TY-1 on viral RNA was evaluated using RT-PCR. As expected, the specific bands disappeared in TY-1-treated FCV and MNV (Figure 5). In addition, the TEM results showed that the treatment with TY-1 induced abnormalities in virion morphology (Figure 6 and Appendix A). These results suggest that TY-1 induces destruction or conformational changes in the capsid protein as well as a disruption in the viral genome. These findings are consistent with the virucidal mechanism of action of allspice oil, which was shown to possibly destroy the MNV capsid and subsequently disrupt the viral genome [45]. Altogether, TY-1 exhibited a similar mode of action against SARS-CoV-2 and IAV [33,34]. The 5.0 mg/mL TY-1 contains 0.034 mg/mL catechins and 0.083 mg/mL TFs. While the contribution of 0.034 mg/mL catechins to IAV inactivation by 5.0 mg/mL TY-1 was limited, that of 0.083 mg/mL TFs was greater [34]. In this study, partial FCV/MNV-inactivation was achieved under treatment with catechins beside TFs treatment (Figure 2A,B), consistent with the previous reports of the virucidal effect of catechin derivatives on FCV and MNV [36,42,44,46]. These results suggest that TFs and catechins both contribute to the virucidal effect of TY-1. While the MNV-inactivating activity of 0.083 mg/mL TFs and 5.0 mg/mL TY-1 was comparable (Figure 2B), the FCV-inactivating activity of TY-1 with this dosage was more potent than the 0.116 mg/mL TFs and catechins (Figure 2A). The result of the experiment targeting FCV in the current study and that targeting IAV in a previous study [34] indicate that TY-1 contains other virucidal components in addition to TFs and catechins. Furthermore, while the use of pure TFs and catechins at virucidal dosage may not be realistic from the aspect of cost, TY-1 has demonstrated a similar or more potent virucidal activity at a much lower cost.

While our findings provide important information regarding the virucidal effect of TY-1 on HuNoV surrogates, the present study has some limitations. Notably, the results obtained from the HuNoV surrogates should be carefully interpreted, as the applicability of the surrogate viruses has been constantly debated and challenged. Specifically, the surrogate viruses do not always behave in the same way as HuNoV [47,48,49]. Hence, two different surrogate viruses, FCV and MNV, which have different characteristics, were used in the present study. Furthermore, comprehensive approaches using multiple experiments testing other HuNoV surrogates (e.g., Tulane virus) could contribute to a better prediction of the virucidal effects of TY-1 against HuNoV. In future studies, the impact of TY-1 on HuNoV itself should also be analyzed. The TEM observation of TY-1-treated HuNoV particles and viability-qPCR [50] may contribute to the evaluation of the impact of TY-1 on virus capsids. Although this study showed the possible application of TY-1 as a disinfectant against HuNoV in solution and on environmental surfaces, the frequent occurrence of food-borne transmissions of HuNoV through the ingestion of raw or partially cooked virus-infected bivalve shellfish should be considered [51]. As there are no currently available disinfectants that can remove HuNoV inside the bivalve shellfish, it may be possible to consider investigating the application of TY-1 in this context. The results reported by Falco et al. [52] suggested the possible synergistic virucidal activity of a gentle heat treatment with plant-derived components such as TY-1, which may contribute to controlling food-borne HuNoV transmission. Overall, further studies aiming to identify whether TY-1 can effectively inactivate HuNoV in bivalve shellfish foods, and on high-touch locations and food contact surfaces, are warranted.

## 4. Materials and Methods

### 4.1. Viruses and Cells

FCV (F9 strain) and Crandell-Rees feline kidney (CRFK) cells were provided by Dr. Ken Maeda at Yamaguchi University, Yamaguchi prefecture, Japan. MNV (S7 strain) was provided by Dr. Yukinobu Tohya at Nihon University, Tokyo, Japan. The murine leukemia macrophage cells (RAW264) were obtained from the RIKEN BRC (Tsukuba, Japan). Both CRFK and RAW264 cells were cultivated in the viral growth medium after FCV and MNV inoculation as previously reported [43]. Purified FCV and MNV solutions were used in some experiments. The viruses were purified by layering the supernatant of the cells with propagated MNV or FCV on 30% sucrose (Nacalai Tesque Inc., Kyoto, Japan) in an ultracentrifuge tube (Hitachi Koki Co., Ltd., Tokyo, Japan) and centrifuged at 100,000× *g* for 3 h. Then, the pellets of viruses were resuspended in PBS.

### 4.2. Preparation of Test Solution Samples

The chemical composition of the TY-1 powder (Yokoyama Food Co., Ltd.; Sapporo, Japan) (Appendix A) [33,34] and the methods of synthesizing and analyzing its corresponding yields, as well preparation of test solutions, were specifically mentioned [34]. In summary, the stock solution of 50.0 mg/mL TY-1 was prepared by dissolving 5.0 g of TY-1 in 100 mL of PBS and collecting the water-soluble layer of TY-1 extract; the resultant solution was stored at −80 °C. Then, solutions with different TY-1 concentrations were prepared by diluting the TY-1 stock solution. Dex represents 50% of TY-1 powder; hence, the 25.0 mg/mL Dex solution was used as a diluent control. This solution was prepared by dissolving 2.5 g of Dex in 100 mL of PBS and storing it at −80 °C. In addition, the 0.067 mg/mL solution of catechin solution was prepared by dissolving 0.067 mg of green-tea-extracted catechin and 5.000 mg of Dex in 1 mL of PBS. In addition, the 0.165 mg/mL solution of TFs was prepared by dissolving 0.112 mg of TF, 0.043 mg of TF-3′-gallate, 0.010 mg of TF-3,3′-digallate, and 5.000 mg of Dex in 1 mL of PBS. Lastly, the 0.232 mg/mL solution of TFs+catechins was prepared by dissolving 0.067 mg of green-tea-extracted catechin, 0.112 mg of TF, 0.043 mg of TF-3′-gallate, 0.010 mg of TF-3,3′-digallate, and 5.000 mg of Dex in 1 mL of PBS. All materials used here were purchased from FUJIFILM Wako Pure Chemical Co. (Osaka, Japan).

### 4.3. Evaluation of FCV and MNV Virucidal Effect of TY-1

The virucidal efficiency of TY-1 against FCV and MNV was evaluated by mixing unpurified viral solutions with an equal volume of 0.6 to 10.0, or 50.0 mg/mL TY-1 or 5.0 or 25.0 mg/mL Dex (solvent control) to achieve the final TY-1 dosages of 0.3 to 5.0, or 25.0 mg/mL and the final Dex dosage of 2.5 or 12.5 mg/mL. The viral solutions were also mixed with an equal volume of PBS. Meanwhile, the virucidal activity of the different components in TY-1 was investigated by mixing 0.067 mg/mL catechins, 0.165 mg/mL TFs, and 0.232 mg/mL solution of TFs+catechins in equal volume with the FCV or MNV solution to achieve the final dosages of 0.034, 0.083, and 0.116 mg/mL, respectively. The viral titers of FCV and MNV in the mixtures were approximately 6.6 and 5.3 log_10_ TCID_50_/mL, respectively. These solutions were incubated at 25 °C for various contact times from 10 s to 24 h, and the viral titers (log_10_ TCID_50_/mL) were calculated [34]. The detection limits of the viral titers in all mixtures were set based on the cytotoxic dosages of the test solutions in CRFK and RAW264 cells under a virus-free condition; the detection limit was 1.3 log_10_ TCID_50_/mL for the solutions containing Dex, PBS, catechins, TFs, TFs+catechins, or 0.3, 0.6, or 1.3 mg/mL TY-1, 2.3 log_10_ TCID_50_/mL for the solutions containing 2.5 or 5.0 mg/mL TY-1, and 3.3 log_10_ TCID_50_/mL for the solutions containing 25.0 mg/mL TY-1.

### 4.4. Evaluation of FCV and MNV Virucidal Effect of TY-1 on a Dry Surface

A volume of 0.06 mL of unpurified FCV and MNV solutions (viral titer: approximately 6.9 and 5.3 log_10_ TCID_50_/mL, respectively) was added to a six-well plate (Nunc, Rochester, NY, USA). After spreading the virus solution over the entire surface of each well and letting it dry completely, 0.1 mL of TY-1 (5.0 mg/mL) or Dex (2.5 mg/mL) was applied over the entire surface of the wells and the plate was incubated at 25 °C for 10 min. Then, the solutions were collected, and the viral titers were evaluated.

### 4.5. Western Blotting

MNV VP1 was evaluated using Western blotting as previously reported [43]. In summary, a purified MNV solution at 5.6 log_10_ TCID_50_/mL was mixed in equal volume with the TY-1 or Dex solution to achieve the final TY-1 and Dex dosages of 5.0 and 2.5 mg/mL, respectively. Next, the mixtures were added to the one-third volume of a 4 × sodium dodecyl sulfate (SDS) sample buffer containing 2-mercaptoethanol (FUJIFILM Wako Pure Chemical Co., Osaka, Japan) instantly (0 h contact time) or after a 24 h incubation at 25 °C (24 h contact time). Then, the mixtures underwent SDS-PAGE, followed by Western blotting analysis to detect the MNV VPI protein using a mouse anti-norovirus (MNV-1) antibody (clone: 5C4.10, Merck & Co., Inc., Kenilworth, NJ, USA) and a goat anti-mouse IgG2b cross-adsorbed secondary antibody, HRP-labeled (Thermo Fisher Scientific Inc. Waltham, MA, USA).

### 4.6. RT-PCR Analysis

The FCV and MNV genes were analyzed using RT-PCR as previously reported [43]. In summary, purified FCV at 7.3 log_10_ TCID_50_/mL and MNV at 7.3 log_10_ TCID_50_/mL were mixed in equal volume with the TY-1 or Dex solutions to achieve the final dosages of TY-1 and Dex at 5.0 and 2.5 mg/mL, respectively. Then, ISOGEN-LS (Nippon Gene Co., Ltd., Tokyo, Japan) was added to the mixtures instantly (0 h contact time) or after a 24 h incubation at 25 °C (24 h contact time). The extracted RNA was reverse transcribed using the FastGene cDNA Synthesis 5 × ReadyMix OdT (NIPPON Genetics Co, Ltd., Tokyo, Japan), and PCR was conducted using the GoTaq^®^ Green Master Mix (Promega Co., Madison, WI, USA) using various primer sequences under specific PCR conditions (Appendix A).

### 4.7. Observation of Morphology of FCV and MNV Virions by TEM

The TY-1 or Dex solution was mixed in equal volume with purified FCV (7.3 log_10_ TCID_50_/mL) or purified MNV (7.3 log_10_ TCID_50_/mL) to achieve the final dosages of 5.0 and 2.5 mg/mL, respectively. The mixtures were incubated at 25 °C for 6 h. Then, negative-stained samples were prepared for TEM analysis as previously described [53]. Lastly, the FCV and MNV particles were observed using TEM (HT7700; Hitachi High-Tech Co., Tokyo, Japan).

### 4.8. Statistical Analysis

The statistical significance of the differences between the TY-1 group with each dosage and the Dex group in each experiment was determined using the Student’s *t*-test. The significance of the differences among the viral titers of the viral solution mixed with TY-1, catechins, TFs, TFs+catechins, or Dex were analyzed using Student’s *t*-test with Bonferroni correction for multiple comparison with the threshold at *p*-value < 0.05. These analysis were performed using Microsoft Excel 2013 (Microsoft Corporation, Redmond, WA, USA).

## 5. Conclusions

The TFs-enriched tea leaf extract TY-1 exhibits virucidal activities against surrogate viruses of HuNoV, one of the most important food-borne viruses. TY-1 exerts a time- and dosage-dependent virucidal effect against the virus solutions. Furthermore, it also inactivates the viruses on a dry surface. TY-1 promotes the profound disruption of virion structures, including the capsid proteins and genome. Therefore, our results demonstrated that TY-1 can be potentially used as a safe virucidal agent that can be applied to viruses both in solution and on environmental surfaces.

## Figures and Tables

**Figure 1 pathogens-11-00533-f001:**
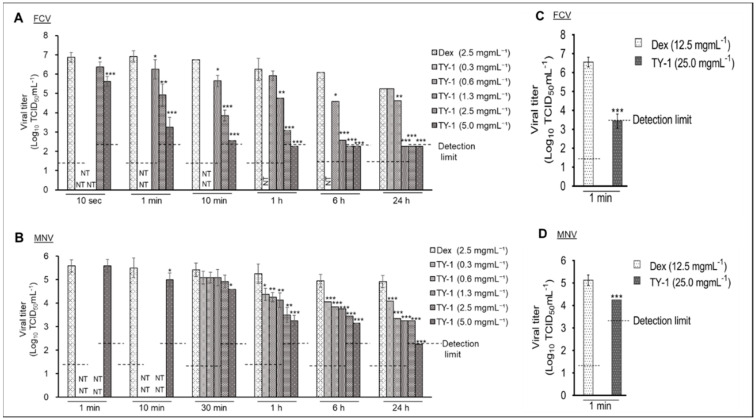
Virucidal effects of TY-1 on FCV and MNV. (**A**–**D**) The FCV (**A**,**C**) or MNV (**B**,**D**) solution was mixed with Dex (final dosage: 2.5 mg/mL (**A**,**B**) or 12.5 mg/mL (**C**,**D**)) or TY-1 (0.3–5.0 mg/mL (**A**,**B**) or 25.0 mg/mL (**C**,**D**)). Then, the mixtures were incubated at 25 °C from 10 s to 24 h. The data are expressed as mean ± SD (n ≥ 6 per group). The Student’s *t*–test was used to analyze the statistically significant differences between the Dex group and TY-1 group with each dosage; * *p* < 0.05; ** *p* < 0.01; *** *p* < 0.001; NT: not tested.

**Figure 2 pathogens-11-00533-f002:**
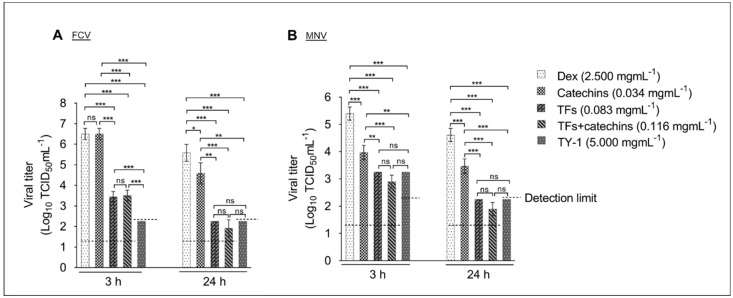
Virucidal effects of catechins, TFs, TFs+catechins, and TY-1 on FCV and MNV. (**A**,**B**) The FCV (**A**) or MNV (**B**) solution was mixed with Dex (final dosage: 2.5 mg/mL), and catechins (0.034 mg/mL), TFs (0.082 mg/mL), TFs+catechins (0.116 mg/mL), or TY-1 (5.0 mg/mL). Then, the mixtures were incubated at 25 °C for 3 and 24 h. The data are expressed as mean ± SD (n ≥ 6 per group). Student’s *t*-test with Bonferroni correction for multiple comparison was used to analyze the statistically significant differences between the different groups; * *p* < 0.05; ** *p* < 0.01; *** *p* < 0.001; ns: not significant.

**Figure 3 pathogens-11-00533-f003:**
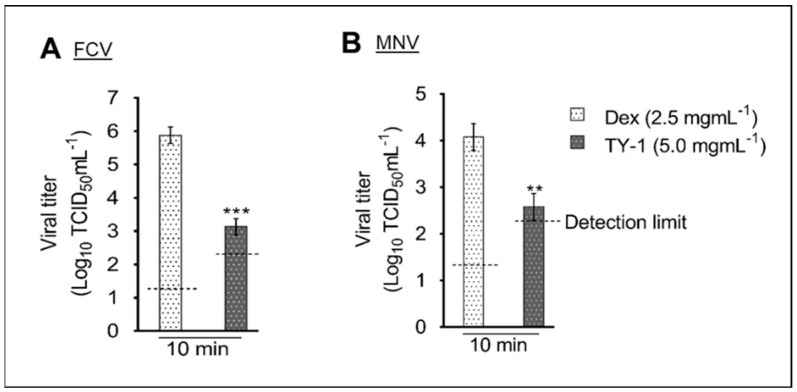
Virucidal effects of TY-1 on FCV and MNV on a dry surface. (**A**,**B**) Dex (final dosage: 2.5 mg/mL) or TY-1 (5.0 mg/mL) was applied on FCV (**A**) or MNV (**B**) on a dry surface, and the samples were placed at 25 °C for 10 min. The data are expressed as mean ± SD (n ≥ 3 per group). Student’s t-test was used to analyze the statistically significant differences between the Dex and TY-1 groups; ** *p* < 0.01; *** *p* < 0.001.

**Figure 4 pathogens-11-00533-f004:**
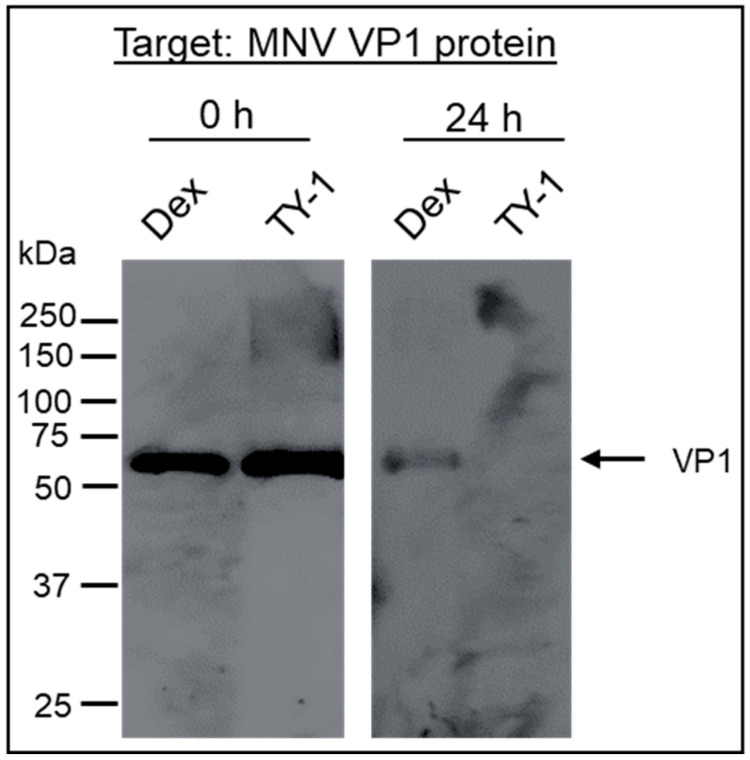
Effect of TY-1 on an MNV structural protein. A solution with purified MNV was mixed with Dex (final dosage: 2.5 mg/mL) or TY-1 (5.0 mg/mL) and incubated at 25 °C for 0 or 24 h. Then, Western blotting was performed to detect the MNV VP1 protein. The results were representative of two individual experiments.

**Figure 5 pathogens-11-00533-f005:**
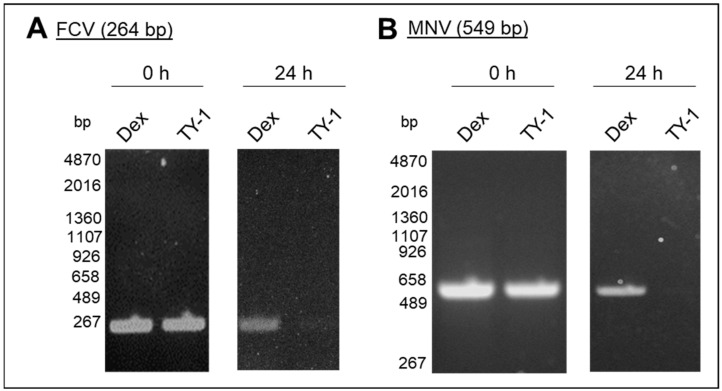
Effect of TY-1 on FCV and MNV genomes. (**A**,**B**) A solution with purified FCV (**A**) or MNV (**B**) was mixed with Dex (final dosage: 2.5 mg/mL) or TY-1 (5.0 mg/mL) and incubated at 25 °C for 0 or 24 h. Then, the extracted viral RNA from the treated viruses was analyzed using RT-PCR. RT-PCR was performed using the FCV-primer set to amplify a 264 bp region on the gene encoding FCV VP1 (**A**) or the MNV-primer set to amplify a 549 bp region on the gene encoding the MNV nonstructural polyprotein (**B**). The results were representative of more than two individual experiments.

**Figure 6 pathogens-11-00533-f006:**
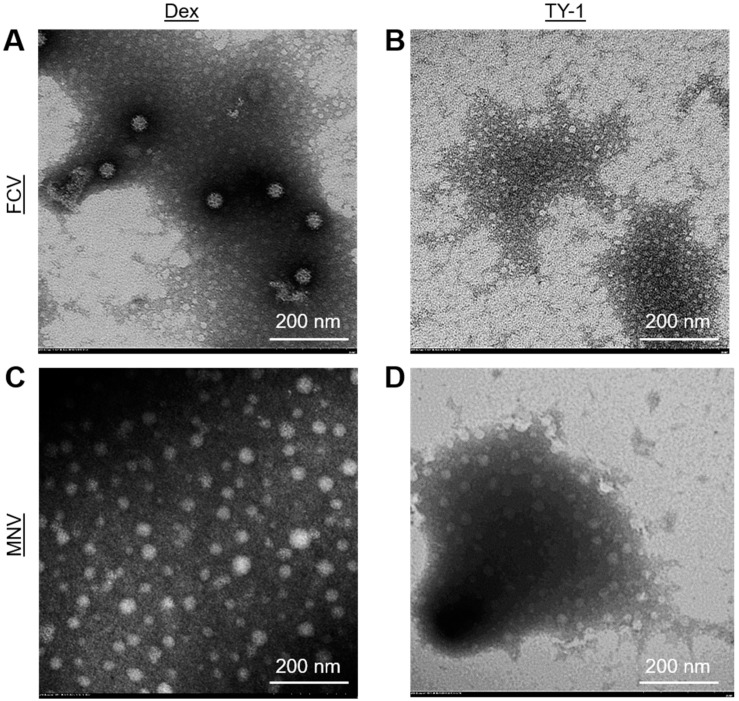
Morphology of Dex- or TY-1-treated FCV and MNV virions. (**A**–**D**) A solution with purified FCV or MNV was mixed with Dex (final dosage: 2.5 mg/mL) (**A**,**C**) or TY-1 (5.0 mg/mL) (**B**,**D**) and incubated at 25 °C for 6 h. Then, the virions were observed using TEM. The results were descriptive images of Dex- and TY-1-treated viral particles.

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
