# Peer review of "Impact of Theaflavins-Enriched Tea Leaf Extract TY-1 against Surrogate Viruses of Human Norovirus: In Vitro Virucidal Study"

_pathogens, 2022, doi:10.3390/pathogens11050533_

Round 1
Reviewer 1 Report
The authors of this paper propose theaflavins-enriched tea leaf extract TY-1 as a potential disinfectant for human noroviruses, testing it on feline calicivirus (FCV) and murine norovirus (MNV) as surrogates. Development of a cost-effective and potent disinfecting agent, which could be safely used when applied directly on food preparation surfaces, would certainly be highly beneficial for the food industry and microbial public health in general. However, it is also necessary to highlight that it would not be able to solve all problems related to foodborne transmission of human noroviruses – there are foods that harbor the virus inside (for example shellfish such as oysters, mussels, clams, etc) and no disinfectant can remove the viruses from these food matrices.
While this work has some interesting aspects, I’m a bit perplexed by the proposed product, as well by a few methodological aspects of this manuscript:
- FCV has been shown to be significantly less resistant compared to other Caliciviruses. While TY-1 seems very efficient against FCV already after one minute, it needs at least 1 hour for some modest reduction of MNV titer (~2 logs) when used at its highest concentration (Fig. 1). This does not seem a lot compared to other disinfectants, and longer exposure times (>30-60 minutes) are often hardly applicable in food industry, therefore I would be concerned about the actual efficacy of TY-1 against human norovirus. Have the authors tested (or considered testing) other, higher concentrations than 5 mg/ml?
- What was the criteria for choosing Dex as a negative control? I believe it would be beneficial to include also another negative control, such as virus incubated in PBS, that gives less reduction of viral titer after 24 hours of incubation (almost 2 logs for FCV). I’m highlighting this also because, in Fig. 3, the VP1 protein band in DEX-treated cells after 24 hours is very weak. Does DEX inactivate the virus as well? Have the authors considered another, less harmful control?
- In Fig. 4, the ladders have very odd sizes – 489, 658, 926 bp…up to 4870 bp. is this so?
- The pdf file “original images” is very confusing. There are either no ladders whatsoever (western blots) or the bands have no description of their size. Also, the different lanes are confusing. For example, in “Target: MNV VP1 protein” – there seems to be 2 very weak bands after 24 hours, and the VP1 (?) band is weaker for Dex compared to TY-1. Also, for the “FCV (264 bp)” PCR images, at 24h I can see 3 bands, maybe 4, but just 2 lanes are described. Also in “MNV (549 bp)” at 24 hours, the 2 bands are distanced one from another, and the sizes are also different. Moreover, in the main manuscript, Fig. 4B omits one band that is visible in the “original images” file (and is not weaker than Dex band…). Could the authors explain why this “original images” file was included (it’s quite messy), and adjust the labeling?
- Fig. 5 is not very convincing (especially 5B, C and D). There seems to be a lot of background (?) and the virions are difficult to distinguish. The white “background” (Fig 5D) seems to be in fact almost virion-like shaped. Do the authors have other images to prove their findings?
- Since there are several other disinfectants described in the literature for noroviruses, why didn’t the authors compare their results to other studies? This seems like a much needed addition to the authors’ discussion.
Author Response
We are grateful to the reviewers for their valuable comments and helpful suggestions, which have allowed us to revise the manuscript appropriately. All changes in the revised manuscript have been indicated by track changes. Below, we have provided our point-by-point responses to the Reviewers’ comments.
Response to Reviewer #1 comments
Point 1. However, it is also necessary to highlight that it would not be able to solve all problems related to foodborne transmission of human noroviruses–there are foods that harbor the virus inside (for example shellfish such as oysters, mussels, clams, etc) and no disinfectant can remove the viruses from these food matrices.
Response 1. We appreciate the Reviewer’s helpful suggestion. We have clarified this point and further elaborated on a future plan to test the virucidal efficacy of TY-1 on such food matrices in the last paragraph of the Discussion section.
Point 2. FCV has been shown to be significantly less resistant compared to other Caliciviruses. While TY-1 seems very efficient against FCV already after one minute, it needs at least 1 hour for some modest reduction of MNV titer (~2 logs) when used at its highest concentration (Fig. 1). This does not seem a lot compared to other disinfectants, and longer exposure times (>30-60 minutes) are often hardly applicable in food industry, therefore I would be concerned about the actual efficacy of TY-1 against human norovirus. Have the authors tested (or considered testing) other, higher concentrations than 5 mg/ml?
Response 2. We have performed additional virucidal experiments using a higher concentration of TY-1 (25.0 mgmL−1) against both FCV and MNV at the reaction time of 1 min. Accordingly, some content have been added in Section 2.1. of the Results, and Figure 1C and D have been added. Further, we have revised the manuscript based on these results.
Point 3. What was the criteria for choosing Dex as a negative control? I believe it would be beneficial to include also another negative control, such as virus incubated in PBS, that gives less reduction of viral titer after 24 hours of incubation (almost 2 logs for FCV). I’m highlighting this also because, in Fig. 3, the VP1 protein band in DEX-treated cells after 24 hours is very weak. Does DEX inactivate the virus as well? Have the authors considered another, less harmful control?
Response 3. Based on the Reviewer’s comment, we have performed an additional experiment to compare the effects of PBS and Dex on the viral titers of FCV and MNV. The results showed that there was no difference between the Dex and PBS groups neither at 0 or 24 h (Figure S1A, B). As shown in Figure 1A, B and Figure S1A, B, the viral titer in the Dex group at 24 h was ~1 log10 TCID50 mL−1 lower than that at 0 h, which explains the weak band densities of VP1 at the reaction time of 24 h in the Dex group (Figure 4).
Point 4. In Fig. 4, the ladders have very odd sizes–489, 658, 926 bp…up to 4870 bp. is this so?
Response 4. We apologize for including incorrect ladder sizes. We have corrected this error (Figure 5).
Point 5. The pdf file “original images” is very confusing. There are either no ladders whatsoever (western blots) or the bands have no description of their size. Also, the different lanes are confusing. For example, in “Target: MNV VP1 protein”–there seems to be 2 very weak bands after 24 hours, and the VP1 (?) band is weaker for Dex compared to TY-1. Also, for the “FCV (264 bp)” PCR images, at 24h I can see 3 bands, maybe 4, but just 2 lanes are described. Also in “MNV (549 bp)” at 24 hours, the 2 bands are distanced one from another, and the sizes are also different. Moreover, in the main manuscript, Fig. 4B omits one band that is visible in the “original images” file (and is not weaker than Dex band…). Could the authors explain why this “original images” file was included (it’s quite messy), and adjust the labeling?
Response 5. We apologize for submitting an incomprehensible “original images” pdf file. This file was submitted based on the submission guidelines of Pathogens. We have revised this file to improve its clarity and have adjusted the labeling per the Reviewer’s suggestion. Specifically, in Figure 4, as we repeated the experiments twice to check the reproducibility of the results, the results of these two experiments have been included in the “original images” file. In Figure 5, we have also repeated the experiments more than twice. In Figure 5B, two different primer sets targeting the MNV genome were used (primer set 1 and 2 amplified 549 and 469 bp PCR products, respectively). Only the result of primer set 1 has been included in the revised manuscript.
Point 6. Fig. 5 is not very convincing (especially 5B, C and D). There seems to be a lot of background (?) and the virions are difficult to distinguish. The white “background” (Fig 5D) seems to be in fact almost virion-like shaped. Do the authors have other images to prove their findings?
Response 6. As TY-1 contains Dex, this background could be related to its effect. The background and viral particles were clearly different. Accordingly, we changed the photo of Figure 6D. In addition, to improve the clarity of our results, we have included Figure S2, which shows different magnification powers.
Point 7. Since there are several other disinfectants described in the literature for noroviruses, why didn’t the authors compare their results to other studies? This seems like a much-needed addition to the authors’ discussion.
Response 7. We would like to thank the Reviewer for this precise suggestion. We have added the related content to the first and second paragraphs of the Discussion section.

Reviewer 2 Report
In the article presented by Mohamed and coworkers the authors investigate the antiviral activity of a theaflavins-enriched extract (TY-1) in the replication of two human norovirus (HuNoV) surrogates, feline calicivirus (FCV) and murine norovirus (MNV). The authors evaluate the ability of the TY-1 to decrease the viral titter calculating the TCID50in cell culture at different concentrations and different times. The authors show that the extract is effective against both viruses, being FCV more sensible to the extract than MNV. Furthermore, the authors investigate how the extract affect the viruses applying different techniques such RT-PCR, western blot and transmission electron microscopy.
Major concerns.
The article is original and properly written but some major issues are observed by this reviewer. In the discussion section it is argued that both surrogates behave different sensitivity to the TY-1 treatment, but it is not discussed that HuNoV may also behave differently to the surrogates. This piece of discussion must be included in the section.
Another weakness of the present job is the incomplete state of the art. It is true that HuNoV has been difficult to grow “in vitro” but due to the differences in viral stability on top of surrogates other strategies have been developed to study HuNoV inactivation. The authors do not mention any of them. For instance, qRT-PCR and viability qRT-PCR have been applied to the study of HuNoV together with their surrogates and to other foodborne viruses such hepatitis A virus. This approach allows a better comparison of the same treatment between culturable and non-culturable viruses. Furthermore, since 2016 it is possible to culture HuNoV in enteroids and this technique has been already utilized to assay the effect of another natural extract, the green tea extract (GTE), against HuNoV. The authors must improve their introduction section including at least the two mentioned alternatives.
Minor concerns.
- Introduction
Lines 30-39. The references 3, 4, 5 and 6 are too old. Please review the literature and utilize more recent references.
Line 56 and following. This could be the place to introduce the use of enteroids to study HuNoV inactivation
- Results
Line 88. Figure 1. Many of the conditions were not tested (NT) please indicate in the text why they were not tested.
Line 88. Figure 1. The reviewer agrees that the proper statistic to be used is the T-Student, but since multiple comparisons are performed with the same data a pValue correction must be done (for instance Bonferroni correction).
Line 107. Figure 2. In this case the authors have decided to apply an ANOVA test instead the T-Student. Since the comparison tries to show that there are differences in the mean value of the TCDI50 with the different treatments the same test as in figure 1 must be used. Please re-analyze the results applying the proper test and its correction.
Line 120. Figure 3. The authors present a western blot of MNV VP1 but the intensity of the bands is not calculated. Please represent graphically the intensity of the bands of the three replicates with the standard deviation and perform statistics to show differences in VP1 integrity. By eye it can be seen that the Dex control has an effect after 24 h. There are several programs that allow band quantification from gels, for instance Image J. Please include the requested analysis.
Line 132. Figure 4. Similarly, to figure 3 the authors do not show the quantification of the bands. Please proceed as indicated above. Alternatively quantitative RT-PCR (RT-qPCR) or viability qPCR should be utilized.
Discussion.
Lines 150 and following. Please see the major concern to this paper and include the discussion indicated.
Line 168. This could be the proper place to discuss a method such as the viability q-PCR to investigate the effect of the YT-1 on HuNoV samples.
Line 178. The sentence “These results indicate that TY-1 was directly degraded the structure of the virion, including external capsid protein and internal viral genome.” Is not properly written. Please write it correctly.
Author Response
We are grateful to the reviewers for their valuable comments and helpful suggestions, which have allowed us to revise the manuscript appropriately. All changes in the revised manuscript have been indicated by track changes. Below, we have provided our point-by-point responses to the Reviewers’ comments.
Response to Reviewer #2 comments
Point 1. The article is original and properly written but some major issues are observed by this reviewer. In the discussion section it is argued that both surrogates behave different sensitivity to the TY-1 treatment, but it is not discussed that HuNoV may also behave differently to the surrogates. This piece of discussion must be included in the section.
Response 1. We have included additional content related to this issue in the last paragraph of the Discussion section.
Point 2. Another weakness of the present job is the incomplete state of the art. It is true that HuNoV has been difficult to grow “in vitro” but due to the differences in viral stability on top of surrogates other strategies have been developed to study HuNoV inactivation. The authors do not mention any of them. For instance, qRT-PCR and viability qRT-PCR have been applied to the study of HuNoV together with their surrogates and to other foodborne viruses such hepatitis A virus. This approach allows a better comparison of the same treatment between culturable and non-culturable viruses. Furthermore, since 2016 it is possible to culture HuNoV in enteroids and this technique has been already utilized to assay the effect of another natural extract, the green tea extract (GTE), against HuNoV. The authors must improve their introduction section including at least the two mentioned alternatives.
Response 2. We have included the aspects highlighted by the Reviewer in the second paragraph of the Introduction section.
Point 3- Introduction:
-Lines 30-39. The references 3, 4, 5 and 6 are too old. Please review the literature and utilize more recent references.
Response 3. We have added more recent references based on the Reviewer’s suggestion and have revised the first paragraph of the Introduction section.
Point 4-Line 56 and following. This could be the place to introduce the use of enteroids to study HuNoV inactivation.
Response 4. We have added content related to this aspect in the second paragraph of the Introduction section.
Point 5. - Results
Line 88. Figure 1. Many of the conditions were not tested (NT) please indicate in the text why they were not tested.
Response 5. We have added an explanation regarding the NT conditions in the Results, Section 2.1.
Point 6. Line 88. Figure 1. The reviewer agrees that the proper statistic to be used is the T-Student, but since multiple comparisons are performed with the same data a pValue correction must be done (for instance Bonferroni correction).
Response 6. In Figure 1A and B, only the statistically significant difference between the Dex group and each test solution group was evaluated. Therefore, a multiple comparison test did not seem necessary here.
Point 7. Line 107. Figure 2. In this case the authors have decided to apply an ANOVA test instead the T-Student. Since the comparison tries to show that there are differences in the mean value of the TCDI50 with the different treatments the same test as in figure 1 must be used. Please re-analyze the results applying the proper test and its correction.
Response 7. In Figure 2, we have changed the statistical analysis based on the Reviewer’s suggestion.
Point 8. Line 120. Figure 3. The authors present a western blot of MNV VP1 but the intensity of the bands is not calculated. Please represent graphically the intensity of the bands of the three replicates with the standard deviation and perform statistics to show differences in VP1 integrity. By eye it can be seen that the Dex control has an effect after 24 h. There are several programs that allow band quantification from gels, for instance Image J. Please include the requested analysis.
Line 132. Figure 4. Similarly, to figure 3 the authors do not show the quantification of the bands. Please proceed as indicated above. Alternatively, quantitative RT-PCR (RT-qPCR) or viability qPCR should be utilized.
Response 8. Due to the short resubmission deadline, we were unable to conduct sufficient additional experiments. Nevertheless, we have already checked the reproducibility of the results. Please check the original raw data pdf file. In relation to the weak band intensity of MNV VP1 in Dex group at 24 h, we have performed an additional experiment to compare the effects of PBS and Dex on the viral titers of FCV and MNV. The results showed that there was no difference between the Dex and PBS groups neither at 0 or 24 h (Figure S1A, B). As shown in Figure 1A, B and Figure S1A, B, the viral titer in the Dex group at 24 h was ~1 log10 TCID50 mL−1 lower than that at 0 h, which explains the weak band densities of VP1 at the reaction time of 24 h in the Dex group (Figure 4).
Point 9. Discussion.
Lines 150 and following. Please see the major concern to this paper and include the discussion indicated.
Response 9. We have added more details and suggestions regarding this point to be considered in potential future studies in the last paragraph of the Discussion section.
Point 10. Line 168. This could be the proper place to discuss a method such as the viability q-PCR to investigate the effect of the YT-1 on HuNoV samples.
Response 10. We have added this aspect in the last paragraph of the Discussion section.
Point 11. Line 178. The sentence “These results indicate that TY-1 was directly degraded the structure of the virion, including external capsid protein and internal viral genome.” Is not properly written. Please write it correctly.
Response 11. We have revised this sentence in the third paragraph of the Discussion section.

Reviewer 3 Report
In their paper entitled “Impact of theaflavins-enriched tea leaf extract TY-1 against surrogate viruses of human norovirus: in vitro virucidal study”, Mohamed report data from the effect of tea leaf extract TY-1 on 2 HuNoV surrogates, FCV and MNV.
The experiments are well-designed, and the paper is well written. However, it would be helpful to describe the limitations of the current study which includes not using the best available surrogates. FCV is a respiratory virus and MNV is also not considered a good surrogate according to a comprehensive paper by Cromeans et al 2014 in Applied Environmental Microbiology. Tulane virus which is an enteric virus in the family Caliciviridae has also been used frequently. Discussing these other surrogate viruses is important so the readers have a better understanding of what is available. In addition, although perhaps not yet practical for many labs, a human intestinal enteroid (HIE) system has been reported by different laboratories and has been used to measure the inactivation of ethanol on HuNoV (Costantini 2018 in EID PMID: 30014841) and also to measure of similar products (green tea) as are described by the authors (Randazzo 2020 in Front Microbiology PMID: 32973702. Also from a practical standpoint, could the authors discuss how TY-1 can be used to reduce HuNoV? I assume by applying TY-1 on surfaces but the authors didn’t show inactivation of FCV and MNV spiked on environmental surfaces.
Author Response
We are grateful to the reviewers for their valuable comments and helpful suggestions, which have allowed us to revise the manuscript appropriately. All changes in the revised manuscript have been indicated by track changes. Below, we have provided our point-by-point responses to the Reviewers’ comments.
Response to Reviewer #3 comments
Point 1. it would be helpful to describe the limitations of the current study which includes not using the best available surrogates. FCV is a respiratory virus and MNV is also not considered a good surrogate according to a comprehensive paper by Cromeans et al 2014 in Applied Environmental Microbiology. Tulane virus which is an enteric virus in the family Caliciviridae has also been used frequently. Discussing these other surrogate viruses is important so the readers have a better understanding of what is available. In addition, although perhaps not yet practical for many labs, a human intestinal enteroid (HIE) system has been reported by different laboratories and has been used to measure the inactivation of ethanol on HuNoV (Costantini 2018 in EID PMID: 30014841) and also to measure of similar products (green tea) as are described by the authors (Randazzo 2020 in Front Microbiology PMID: 32973702.
Response 1. We highly appreciate the Reviewer’s valuable comments. We have added detailed descriptions related to the cultivation of HuNoV and the limitations of using surrogate viruses in the Introduction and Discussion sections.
Point 2. Also, from a practical standpoint, could the authors discuss how TY-1 can be used to reduce HuNoV? I assume by applying TY-1 on surfaces but the authors didn’t show inactivation of FCV and MNV spiked on environmental surfaces.
Response 2. We highly appreciate the Reviewer’s valuable suggestion. To further investigate and clarify this question, we have performed an additional experiment to evaluate the virucidal activities of TY-1 against FCV and MNV on dry surfaces. The detailed description of the experimental design and results have been included in the Materials and Methods, Section 4.4.; Results, Section 2.2.; and Figure 3.
Reviewer 4 Report
The manuscript entitled “Impact of theaflavins-enriched tea leaf extract TY-1 against surrogate virus of human norovirus: in vitro virucidal study” investigates the virucidal activity of natural substances against food borne viral pathogens. The manuscript is well written and provides new information on the viral inactivation of natural substance in alternative to chemical compounds. I have no changes to highlight, so the manuscript can be accepted in present form.
Author Response
We are grateful to the reviewers for their valuable comments and helpful suggestions, which have allowed us to revise the manuscript appropriately. All changes in the revised manuscript have been indicated by track changes. Below, we have provided our point-by-point responses to the Reviewers’ comments.
Response to Reviewer #4
The manuscript entitled “Impact of theaflavins-enriched tea leaf extract TY-1 against surrogate virus of human norovirus: in vitro virucidal study” investigates the virucidal activity of natural substances against food borne viral pathogens. The manuscript is well written and provides new information on the viral inactivation of natural substance in alternative to chemical compounds. I have no changes to highlight, so the manuscript can be accepted in present form.
Response: We highly appreciate and are thankful for the Reviewer’s opinion and acceptance of the manuscript.
Round 2
Reviewer 2 Report
The authors have answered properly to all the concerns raised, so the paper can be now accepted.